# Part2Point: A Part-Oriented Point Cloud Reconstruction Framework

**DOI:** 10.3390/s24010034

**Published:** 2023-12-20

**Authors:** Yu-Cheng Feng, Sheng-Yun Zeng, Tyng-Yeu Liang

**Affiliations:** Department of Electrical Engineering, National Kaohsiung University of Science and Technology, No. 415, Jiangong Road, Sanmin District, Kaohsiung City 807618, Taiwan; f110154123@nkust.edu.tw (Y.-C.F.); f111154152@nkust.edu.tw (S.-Y.Z.)

**Keywords:** 3D modeling, artificial intelligence, point cloud, part segmentation, high resolution, parameter amount

## Abstract

Three-dimensional object modeling is necessary for developing virtual and augmented reality applications. Traditionally, application engineers must manually use art software to edit object shapes or exploit LIDAR to scan physical objects for constructing 3D models. This is very time-consuming and costly work. Fortunately, GPU recently provided a cost-effective solution for massive data computation. With GPU support, many studies have proposed 3D model generators based on different learning architectures, which can automatically convert 2D object pictures into 3D object models with good performance. However, as the demand for model resolution increases, the required computing time and memory space increase as significantly as the parameters of the learning architecture, which seriously degrades the efficiency of 3D model construction and the feasibility of resolution improvement. To resolve this problem, this paper proposes a part-oriented point cloud reconstruction framework called Part2Point. This framework segments the object’s parts, reconstructs the point cloud for individual object parts, and combines the part point clouds into the complete object point cloud. Therefore, it can reduce the number of learning network parameters at the exact resolution, effectively minimizing the calculation time cost and the required memory space. Moreover, it can improve the resolution of the reconstructed point cloud so that the reconstructed model can present more details of object parts.

## 1. Introduction

In recent years, 3D modeling has become increasingly widespread in virtual reality (VR), augmented reality (AR), and computer vision. These applications enrich people’s lives and bring much convenience. For example, in the realm of gaming [1], it can offer players a fantastic and realistic gaming experience. In architecture and real estate, it can be used to create virtual models of buildings [2], enabling designers and clients to visualize better and understand the appearance and structure of the buildings. In the medical field, it can be used to create precise models of human organs, assisting doctors in surgical planning and simulation [3]. In education and training [4], it can be used to establish virtual laboratories, simulated environments, and interactive learning tools to help students better comprehend complex concepts and operations. In the entertainment and film industry [5], it can be employed to create realistic special effects in animations and making movies. It can be foreseen that the applications of 3D modeling will continue to expand and penetrate people’s daily lives.

However, creating 3D models took much work in the past. Application developers had to manually use computer-aided design software such as Blender [6], Solidwork [7], and ProE [8] to model objects one by one according to their structures. To save labor and time, an alternative method is to scan physical objects using 3D scanners directly. However, the scanning range of 3D scanners is limited, and high-resolution 3D scanner equipment is expensive, making it difficult for the general public to afford. By contrast, structure-from-motion (SfM) photogrammetry [9,10,11] is a practical and realistic solution to create three-dimensional models of physical objects from a collection of two-dimensional images or video frames. The steps involved in SfM mainly include feature detection and matching, camera pose estimation, triangulation, and point cloud generation. SfM is a well-established method with simplicity and versatility, while it is sensitive to outliers in feature matching and relies on dense and distinctive features for accurate reconstruction. It is necessary to seek or integrate with an alternative approach for generating promising 3D reconstructions, even with sparse input data, and handling low-texture and -occlusion environments more effectively by learning contextual information.

Fortunately, the rapid development of graphics processing units (GPUs) in recent years has led to artificial intelligence (AI) breakthroughs. Along with the release of large open-source 3D object databases such as ShapeNet [12], Pix3D [13], and ModelNet [14], deep learning has gained excellent development opportunities in the 3D modeling field. By inputting multiview 2D images of objects, machines can learn various features of objects and automatically reconstruct the 3D models of the entities. These models can be used to create 3D digital content later. This dramatically accelerates the development of 3D modeling applications, making AI-based 3D modeling a popular trend in research. The primary methods for representing 3D object models are voxels [15], meshes [16], and point clouds [17]. Point clouds require less data than voxels and are suitable for handling large-scale models. They can directly capture and represent real-world data, such as 3D scans. They also can achieve a similar level of detail as meshes by sampling specific points on the object’s surface while avoiding using complex polygonal data structures to describe an object. Considering resolution and computational costs, this study adopts point clouds as the target for 3D model reconstruction.

Past research has proposed numerous 2D-to-3D point cloud generators. Their architecture can be divided into two parts. The first part is the encoder, responsible for extracting latent features from input 2D images. The second part is the decoder, which transforms the obtained elements into a 3D point cloud. The overall architecture resembles an autoencoder. With the continuous evolution of encoder and decoder architectures, the performance of reconstructing 3D point clouds through autoencoders has reached practical levels. However, most models currently maintain resolutions of around 1024 or 2048 points. For particular complex objects, this can lead to losing many details in the object’s shape and reducing the model’s realism. As shown in Figure 1, the point cloud comparison of the bicycle at different resolutions is illustrated. The point clouds are represented using 16,384, 8192, and 2048 points, respectively. The details of the bicycle’s wheels, rear rack, and handlebars are visible at the highest resolution, while some accessory details are missing around the handlebars at the lower resolution. More information including the wheel spokes, rear rack, and handlebars is lost at the lowest resolution, since directly increasing the output resolution of the model would substantially inflate the number of parameters in 3D modeling. Consequently, the model’s training time and memory requirements would also dramatically increase. Moreover, the point cloud generators proposed by past studies often lack constraints on local structures while learning the construction of an object model. This leads to a lack of diversity in the generated 3D objects. Consequently, the local structure of the object needs to be accurately reconstructed.

As previously described, this study aims to increase the output resolution of deep learning models for point cloud reconstruction while not increasing the number of model parameters. Inspired by the divide and conquer approach, we propose a part-oriented point cloud reconstruction framework called Part2Point. The key feature of this framework is its process of performing 2D part segmentation on the input 2D image data. This segmentation divides the reconstruction target into smaller parts, initially a complete object. These parts are then sequentially inputted into the point cloud generator to reconstruct corresponding part point clouds. Finally, these individual part point clouds are merged to generate a complete point cloud. On the other hand, this framework uses gamma adjustment to enhance the part image features to extract the part image with better features in the image processing stage. Moreover, it calculates the loss values of the reconstruction results by comparing them with the actual part clouds segmented from the 3D ground-truth point cloud to optimize the output of the 3D part point clouds reconstructed from 2D part images.

The remaining sections of this paper are organized as follows: Section 2 introduces related research, and Section 3 explains the framework of Part2Point. Section 4 presents and discusses the experimental results of performance evaluations. Finally, we give the conclusions of this paper and future work.

## 2. Related Work

The proposed framework involves many research topics related to point clouds, such as feature extraction, reconstruction, localized comprehension, and semantic segmentation. The related work about these topics is described as follows.

Feature extraction: PointNet [18] introduced a method for feature learning and classification of irregular and unordered 3D point clouds. It utilizes multilayer perceptrons (MLPs) and max pooling in the local feature extraction network, and fully connected layers and max pooling in the global feature extraction network. PointNet++ [19] employs a hierarchical structure to capture local and global features in 3D point clouds. It divides the point cloud into multiple levels of regions, progressively extracting features from small local areas to the entire global part of the point cloud. Each layer uses the PointNet method to enhance understanding of local features. To address incomplete point clouds, which can hinder essential feature extraction, PCN [20] proposes a point cloud completion method based on autoencoders. It predicts missing points from incomplete 3D point clouds to restore the missing parts. PU-Net [21] learns point cloud features through PointNet and utilizes multibranch convolution units for feature transformation and deconvolution operations, converting sparse 3D point clouds into denser ones. Zhang et al. [22] introduced a self-supervised method for upsampling sparse point clouds to increase their density. N. Engel et al. [23] proposed a deep learning network called the point transformer to extract local and global features and relate both representations by the local–global attention mechanism to capture spatial point relations and shape information. This network is permutation-invariant because of a module called SortNet that extracts ordered local feature sets from different subspaces. M.H. Guo et al. [24] presented the point cloud transformer (PCT) to capture better local features hidden in the point cloud by the offset-attention of the implicit Laplace operator and the normalization mechanism that is inherently permutation-invariant and more effective than the original transformer for learning point clouds. H. Zhao et al. [25] constructed a 3D point cloud understanding network based on point transformer layers, pointwise transformations, and pooling.

Reconstruction: Several approaches have been developed for reconstructing 3D point clouds from 2D single-view images. PSGN [26] introduced a framework and loss formula for this purpose. Lin et al. [27] synthesized depth images and used supervised depth images to enhance point cloud reconstruction. Jin et al. [28] extended [27] using weak supervision to generate depth images for improved reconstruction. Mandikal et al. [29] predicted a low-resolution point cloud from a 2D image and upsampled it to reconstruct a high-resolution point cloud. DeformNet [30] retrieved a point cloud shape template and fused it with image features. 3D-LMNet [31] employed a multistage approach with an autoencoder architecture. Pix2Pc [32] used VGG19 to extract features and then reconstructed the point cloud through a deformation network. 3D-ReConstnet [33] proposed an end-to-end 3D reconstruction network based on a variational autoencoder (VAE). Pixel2point [34] used CNN to extract image features and combine them with a spherical template. Yuniarti et al. [35] extended templates to more categories, while Ping et al. [36] projected point clouds onto a 2D plane and used Gaussian derivatives and Harris corner detection to reduce differences. Liu et al. [37] employed multiscale features and random point clouds for global feature generation. Chen et al. [38] used image retrieval and filtering of object regions to reconstruct point clouds. 3D-SSRecNet [39] introduced a point cloud reconstruction network based on the DetNet backbone and ELU activation function, providing a higher receptive field for feature extraction.

Localized comprehension: Prior research in point cloud understanding and reconstruction primarily focused on learning overall object features directly for reconstruction. However, some studies emphasized understanding object composition for capturing local structures in reconstructed point clouds. Yu et al. [40] treated 3D part segmentation as a multiclass labeling problem, introducing a top-down recursive decomposition method. They recursively decomposed 3D models into binary nodes, using a classifier to determine decomposition type and stopping criteria, with leaf nodes representing 3D part segmentation results. Ko et al. [41] presented a model generating 3D point clouds with semantic parts. Using latent 3D model features, the generator expanded the point cloud iteratively in a coarse-to-fine manner, starting from a single point. Wang et al. [42] proposed the part tree to point cloud (PT2PC) model based on conditional generative adversarial networks (GANs). They traversed the input part tree, extracting subtree features bottom-up and recursively decoding part features top-down to generate the final part point cloud. Li et al. [43] used a variational autoencoder (VAE) to obtain global latent features mapped into three different features and concatenated for reconstructing the complete point cloud and primitives. Niu et al. [44] introduced the structure recovery network, mapping input images directly to a hierarchical structure representing part structures for recovering part structures from a single 2D view. Mandikal et al. [45] proposed an architecture combining point cloud reconstruction networks with segmentation networks. They used a novel loss function called location-aware segmentation loss to enable information sharing between the two networks, optimizing the output point cloud model.

Semantic segmentation: In single-view 3D reconstruction tasks, the input 2D images significantly impact the model’s output, making proper preprocessing of input data necessary. Liu et al. [46] introduced a pixel-level classification approach for street view images, dividing them into four semantic and depth regions. Cordts et al. [47] employed fully convolutional networks (FCN) for pixel and instance-level segmentation in complex urban scenes to capture spatial information effectively. DeepLabv3 [48] utilized residual networks (ResNet) for feature learning and multiscale dilated convolution to capture features of various scales for scene-level segmentation. Zhou et al. [49] analyzed the ADE20K dataset, which covers scenes, objects, and parts, and tested various semantic segmentation models. They need a hierarchical segmentation module for handling different segmentation levels. Liu et al. [50] addressed the challenges of manual part segmentation by annotating 3D CAD models and rendering numerous synthetic 2D images from these models. Atik et al. [51] present an effective ensemble deep learning method for semantically segmenting mobile LiDAR sensor point clouds. The approach involves projecting 3D mobile point clouds into a 2D representation using spherical projection on range images. Liu et al. [52] introduced the spatial eight-quadrant kernel convolution (SEQKC) algorithm for enhancing 3D point cloud semantic segmentation, specifically targeting small objects in complex environments. The SEQKC algorithm improves the network’s capability to extract detailed features, resulting in higher accuracy for small objects and boundary features.

In summary, previous methods primarily relied on 2D image data to reconstruct global point clouds directly. However, they faced limitations in generating high-resolution point clouds, as achieving higher resolution necessitates increasing the decoder’s output neuron count. While Mandikal et al.’s [29] method can generate higher-resolution point clouds, it initially reconstructs sparse point clouds and incrementally increases resolution, potentially losing object details similar to upscaling a compressed 2D image. 3D-PSRNet’s [45] reconstruction method incorporates object parts into network training, allowing the segmentation network to capture detailed object part features. However, this approach requires training in both reconstruction and segmentation networks, leading to increased model parameters and similar limitations on point cloud resolution output.

By contrast, our study utilizes 2D and 3D part annotation data during model training to address these challenges to enhance output resolution, retain object details, avoid increasing model parameters, and maintain object detail comprehension. We employ the 2D annotation image from Liu et al. [50] based on the ShapeNet [12] dataset for 2D part segmentation training, along with ShapeNet part annotation [53] point cloud annotation data. We also use the DeepLabV3 method proposed by Zhou et al. [48] to train the part segmentation model. The comparison of the related studies with Part2Point is summarized in Table 1.

## 3. Proposed Framework

This paper proposes a part-oriented point cloud generation framework called Part2Point, as shown in Figure 2. The process of reconstructing a point cloud model for a given object image is shown in the (a) block. Firstly, the input object image undergoes data preprocessing and is segmented into several part images, and each only contains some local structure of the object. Then, the part images are individually input into the point-cloud generator to reconstruct the part point clouds of the object. Finally, the part point clouds are combined to form the complete object point cloud. This architecture allows the neural network to focus on learning the reconstruction of local structures of objects while reducing the number of parameters in the generator model, preventing an explosion in the parameter number with increasing overall point cloud resolution. On the other hand, the model’s training process is as outlined in block (b). The first is to input the 2D images (*x*) of objects, segmenting each 2D object image into x-part images ($x_{part}$). The second is to input the x part images individually into the point cloud generator for producing predicted part point clouds. The third is to calculate the part loss ($L_{part}$) by comparing the predicted part point clouds ($y_{pred}$) with the ground truth ($y_{part}$) of the part point clouds. The fourth is to merge the predicted part point clouds to form a complete object point cloud ($y_{output}$) and compare it with the 3D ground truth (*y*) of the object point cloud to calculate the global loss ($L_{global}$). The final is to optimize the parameters of the point-cloud generator according to the local and global losses.

As previously described, the stages of point cloud reconstruction in the proposed framework are mathematically expressed as follows.

Stage 1: $x_{part}=S_{2D}\left(x\right)$, where $S_{2D}\left(x\right)$ is a function to separate an input image $\left(x\right)$ into a set of part images denoted as $x_{part}$.

Stage 2: $y_{pred}=G\left(x_{part}\right),$ where $G\left(x_{part}\right)$ is a function to transfer $x_{part}$ into a set of part point clouds denoted as $y_{pred}$.

Stage 3: $y_{output}=M\left(y_{pred}\right),$ where $M\left(y_{pred}\right)$ is a function to merge $y_{pred}$ into an output point cloud denoted as $y_{output}.$

Stage 4: $y_{part}=S_{3D}\left(y\right),$ where $S_{3D}\left(y\right)$ is a function to separate the 3D ground truth $\left(y\right)$ of the input image into a set of part point clouds denoted as $y_{part}$.

Stage 5: Compute the values of $L_{part}\left(y_{pred},y_{part}\right)$ and $L_{global}\left(y_{output},y\right)$, where $L_{part}\left(y_{pred},y_{part}\right)$ is a loss function to estimate the difference between $y_{pred}$ and $y_{part};$ $L_{global}\left(y_{output},y\right)$ is a loss function to estimate the difference between $y_{output}$ and y.

Stage 6: Optimize the point cloud generator (i.e., the $G\left(x_{part}\right)$ function) according to the values of $L_{part}\left(y_{pred},y_{part}\right)$ and $L_{global}\left(y_{output},y\right)$.

Stages 4–6 are performed only when training the point cloud generators, and 3D part segmentation is executed in advance to save training time. The following are the details of the part segmentation, point cloud generator, and loss functions in the Part2Point framework.

### 3.1. Part Segmentation

The 3D part segmentation denoted as $S_{3D}$, assigns a class label to each point, essentially classifying points into different parts. To calculate reconstruction errors for individual part point clouds in Part2Point, the ground-truth point cloud is segmented into multiple parts, where each part point cloud corresponds to a specific part of the object. For example, in the “airplane” category, the segmentation might include part point clouds for the fuselage, tail, and wings (as shown in Figure 3). However, due to variations in the resolutions of segmented part point clouds, batch training during model training is impractical. We draw inspiration from PointNet++ [19] to address this challenge and perform uniform point subsampling on the input point clouds. We utilize the farthest point sampling (FPS) technique to subsample the segmented point clouds to a consistent resolution.

Similarly, the 2D part segmentation denoted as $S_{2D}$ assigns a class label to each pixel in the image. The image is divided into distinct parts, with each containing the pixels of a specific part area, while other regions are masked and filled with black color. This segregation enables the extraction of unique features for each part to facilitate point cloud reconstruction. However, challenges arise when objects are occluded from certain viewpoints, causing some part images to be unavailable. For instance, the wheels may be occluded when viewing a car from directly above, causing the 2D part segmentation to miss the wheel parts. To address this issue, we utilize the original input image to fill in the missing part images (as depicted in Figure 4). We currently employ the DeepLab [54] architecture to train a neural network for 2D part segmentation to annotate part information in 2D images.

### 3.2. Point Cloud Generator

As shown in Figure 2, the role of the point cloud generator G is to reconstruct the predicted part point cloud $y_{pred}$ from independent part images $x_{part}$ using a neural network, defined as $y_{pred}=G\left(x_{part}\right)$. $N_{pred}$ represents the resolution of the predicted part point cloud. Once the reconstruction of individual object parts is completed, a loss function calculates the error between the predicted part point cloud and the actual part point cloud $y_{part}$, defined as $L_{part}\left(y_{pred},y_{part}\right)$. On the other hand, after reconstructing the predicted part point clouds, they are merged to obtain the model’s output point cloud $y_{output}$, which is then compared to the actual point cloud to calculate the global loss, defined as $L_{global}\left(y_{output},y\right)$. The loss function of the neural network is defined as $L=L_{part}+L_{global}$.

For merging part point clouds, the order of points does not need to be processed individually due to the permutation invariance property of data structure. It is sufficient to reshape the tensor size from $\left(X^{\prime },N_{pred},3\right)$ to $\left(X^{\prime }\times N_{pred},3\right)$, where the segmented point cloud parts are sequentially arranged as a single point cloud. This operation is defined as $y_{output}=M\left(y_{pred}\right)$, as shown in Figure 5.

### 3.3. Loss Function

The loss function evaluates the error between the output and the ground truth. It is a crucial factor affecting the learning performance of neural networks, as the network optimizes and updates its weights based on this error. In the point cloud generation task, the loss function measures the distance between the generated and real point clouds. Since a point cloud is an unordered set of points, the meaning remains the same regardless of the order of the points. Currently, the loss functions for point cloud reconstruction usually are the chamfer distance (CD) [55] and Earth mover’s distance (EMD) [56]. The chamfer distance is defined as Equation (1).
(1)$$L_{CD}\left(S_{1},S_{2}\right)=\sum_{x\in S_{1}}\underset{y\in S_{2}}{\min}{\left\|x-y\right\|}_{2}^{2}+\sum_{y\in S_{2}}\underset{x\in S_{1}}{\min}{\left\|x-y\right\|}_{2}^{2}$$

Past studies [33,34,39] used CD as the loss function due to its computational efficiency. However, these studies found that CD could not generate high-quality point clouds with uniform distribution. This often led to the clustering or splattering of points in the reconstructed output, resulting in a visual effect that was not satisfactory.

On the other hand, the Earth mover’s distance is defined as Equation (2).
(2)$$L_{\mathrm{EMD}}\left(S_{1},S_{2}\right)=\underset{\varphi :S_{1}\to S_{2}}{\min}\sum_{x\in S_{1}}{\left\|x-\varphi \left(x\right)\right\|}_{2}$$

In this equation, φ is a one-to-one mapping function that maps $S_{1}$ to $S_{2}$. Due to the one-to-one mapping relationship, EMD does not have the clustering characteristics existing in CD. The drawback of EMD is that it involves enormous computational complexity for high-resolution point clouds. Therefore, in most existing methods, the EMD algorithm is approximated to compute the point cloud error.

As shown in Figure 6, the evaluation results of different reconstruction outputs using CD and EMD show that CD’s evaluation does not accurately capture the actual visual effect of the objects. Although the CD metric considers Output2 to have a minor error compared to Output1, the points in Output2 are excessively concentrated in some regions and need to accurately match the points of the actual object, resulting in unclear details. In contrast, EMD requires finding a mapping function that ensures each point in the output point cloud is mapped to a unique point in the actual point cloud. This ensures that the output point cloud has the same distribution and density as the true point cloud, making it more capable of capturing local details and density distribution. Although CD has inherent drawbacks, its efficient computation can assist point cloud reconstruction networks in faster convergence during training. By using additional constraints to aid model training, this study uses EMD and CD to compute $L_{part}$ and $L_{global}$ when training the point cloud generator.

## 4. Experimental Results

We have evaluated the performance of Part2Point in this paper. We used three point cloud generators, namely 3D-ReconstNet, Pixel2point, and 3D-SSRecNet, in the Part2Point framework. We evaluated these generators without and with part segmentation to assess differences in model reconstruction performance, model parameters, training time, and point cloud resolution. We also tested the impact of the additional global loss function and part numbers on the Part2Point reconstruction performance. Our experimental environment set up for this performance evaluation included a PC with an Intel i9-10980XE CPU, an NVIDIA GeForce RTX 3090 Ti graphic card, 256 GB RAM, and a Linux Ubuntu 20.04.5 operating system. We used Python programming language, CUDA 12.0 library, and the Pytorch [57] deep learning framework for developing the test program and employed CloudCompare [58] for point cloud visualization and rendering of point cloud part segmentation. During the training of these generators, the input image size was set to 128 × 128, and the used optimizer was Adam, with a learning rate of 5 × 10^−5^. Each batch had 32 data samples, and the training lasted 50 epochs.

On the other hand, the datasets used in the tables and figures of experimental results are UDA-Part [50] and ShapeNet Part Annotation [53]. We utilized the multiview 2D part segmentation dataset provided by UDA-Part and extracted corresponding 3D part segmentation point clouds from the ShapeNet part annotation as reconstruction targets for the 2D dataset. The dataset categories include cars, airplanes, bicycles, buses, and motorcycles. For the experiments, 80% of the data were used for training the used point cloud generators, while the remaining 20% served as the test set. However, the existing ShapeNet part annotation dataset only contains point cloud objects with a resolution of approximately 2000 points for representation. To test the performance of point cloud reconstruction at higher resolutions, we used the point cloud processing software, CloudCompare [58], to resample 15,000 points on the mesh surface for each object from ShapeNetCore, where each point was annotated with part information. While testing the performance of reconstructing point clouds, we calculated the distance error between each part of the generated point cloud and its corresponding ground truth. Then, we summed and averaged the errors belonging to the same object part to provide the overall error of the output point cloud. As discussed in Section 3.3, we used the EMD metric for the error of point cloud reconstruction in our experimental results.

### 4.1. Impact of Part Segmentation

This experiment assessed the influence of part segmentation on both the computational cost and effectiveness of point cloud generators at different resolutions (1536, 4608, 9216, and 13,824). Firstly, we present the impact on execution cost in Table 2.

Obviously, the parameters of the point-cloud generators with part segmentation are much less than those of the same generators without part segmentation. When part segmentation is utilized, the point cloud generators individually reconstruct point clouds for the part segments of a given object and then assemble the point clouds of part segments to form the complete point cloud of the object. For the same resolution, as the point cloud is divided into three parts, the number of output points of the decoder is only one-third of that without part segmentation. Consequently, the parameters of the generator models decrease significantly. However, when an object undergoes three part segmentations, the generator must create three part point clouds for each object and combine them into a complete point cloud. When the resolution is small (e.g., 1536), the increased time for generating more part point clouds may offset the time saved by reducing generator parameters, resulting in an overall increase in point cloud generation time for some cases, such as 3D-ReconstNet and 3D-SSRecNet. However, as the resolution increases, the parameter reduction begins to outweigh the effects of increased part point generation, leading to a significant decrease in batch time for all generators. Notably, at resolutions 9216 and 13,824, all three generators cannot execute without part segmentation due to high memory demands exceeding the graphics card capacity. However, with part segmentation, the model parameters do not excessively increase with resolution, enabling the generators to effectively generate point clouds regardless of the resolution.

Table 3 presents the Earth mover’s distance (EMD) errors for point cloud reconstruction with three parts and resolutions of 1536, 4608, 9216, and 13,824. The results consistently show that employing part segmentation improves the EMD metric for point cloud reconstruction. This experiment highlights that integrating part segmentation into point cloud reconstruction tasks can enhance reconstruction results at most resolutions. Although the parameter number of the generator model is much less than the original, the generators with part segmentation can reduce the EMD values because they construct better point clouds for each part of the target object as they can focus on local part features. With increasing resolution, the model parameters also grow, enabling the model to capture finer object details, theoretically resulting in improved point cloud quality. However, the reconstruction results induced by 3D-ReconstNet and Pixel2point are not improved by part segmentation at the 1536 resolution. This result implies that although part segmentation can effectively reduce the model parameters necessary for reconstructing point clouds, the reconstruction accuracy may not be improved but degraded when the parameter number is reduced below a threshold. In addition, the improvement induced by part segmentation seems most pronounced when transitioning from resolutions 1536 to 4608. The possible explanation could be that the current model architectures for the generators are not yet optimized for higher resolutions like 9216 and 13,824. Future work may require adjustments to the generator model architecture to achieve better reconstruction quality with more points.

Figure 7, Figure 8, Figure 9 and Figure 10 show the visualizations of point clouds generated by 3D-SSRecNet with and without part segmentation. In each figure, from left to right, we have the input image, image-based part segmentation, point cloud generated without part segmentation (w/o part), point cloud generated with part segmentation (w part), and the ground-truth point cloud. At a resolution of 1536, each part contains only 512 points. The resolution of part point clouds is too low to show the visual differences in the reconstructed point clouds.

At a resolution of 4608, the ground-truth point clouds exhibit more detailed object information. When applying part segmentation, the reconstructed point clouds are closer to the real ones. For instance, on the bicycle, the frame’s support structure in the wheels is reconstructed more precisely and uniformly. The separation between the rear mudguard and the tire is more precise, resulting in a more uniform and smoother appearance overall. Conversely, without part segmentation, the reconstruction of the wheel’s support structure and the rear mudguard appears blurred and exhibits irregularities and roughness. Even at resolutions of 9216 and 13,824, the point cloud generators with part segmentation still achieve nearly realistic point cloud reconstructions. For example, on the bicycle, the brake lines on the frame and the chain on the derailleur are reconstructed with fine details, and the overall contour lines are denser and more distinct. In contrast, without part segmentation, the point cloud generator lacks the memory capacity to adequately train and reconstruct point clouds.

### 4.2. Impact of Part Number

This experiment aims to assess the impact of the number of parts on the proposed framework’s performance. We measured model parameters, training time, and point cloud generator reconstruction errors for configurations with two and three parts. The results are summarized in Table 4 and Table 5. When maintaining the same resolution, increasing the number of parts substantially reduces the generator’s model parameters. Additionally, we observed that the training time per batch increased with resolution. However, using a generator with more parts actually results in less training time. This phenomenon occurs because as the number of parts grows, the point cloud can be divided into smaller part clouds, reducing the reconstruction time for each part. The time saved in generating additional parts outweighs the time spent, leading to an overall reduction in training time. This effect may not be very noticeable at lower resolutions but becomes increasingly pronounced as the resolution rises. Conversely, increasing the number of parts leads to a higher overall point count in the generated objects.

Table 5 presents the Earth mover’s distance (EMD) evaluation for point cloud reconstruction using two and three parts. The table reveals that increasing the number of part point clouds results in improved EMD values. These improvements become more significant as the resolution increases. Part segmentation enables the generator to focus on the distinct features of each part. As the resolution rises, the generator’s model parameters also increase, enhancing its capacity to capture finer details of object parts.

### 4.3. Impact of Global Loss

This experiment aims to conduct investigations and explore the impact of using an additional global loss function to constrain the global point cloud in terms of Part2Point’s performance. Table 6 shows that after using the additional global loss function, the EMD errors of the generated reconstruction models improve regardless of the resolution or the type of generator used. This indicates that incorporating an extra global loss function effectively aids in merging and fusing the part point clouds into a coherent whole object. This result suggests that the global loss function can capture the overall features and structure of the entire point cloud, leading to more consistent and natural-looking generated point clouds. Consequently, this method shows significant effectiveness in enhancing the performance of the part point cloud generator. The experimental findings may contribute to further improvements in point cloud generation by achieving the generated point clouds’ global and shape consistency.

### 4.4. Reconstruction Errors of Different Object Categories

Table 7 presents the EMD values of point clouds generated for various object categories when the number of part segments is 3. At low resolutions, the three generators show improved EMD values in only 1 or 2 categories. As the resolution increases, the model parameter also rises, enhancing the point cloud generators’ ability to learn object details. At a resolution of 4608, EMD values for three categories are improved in the point clouds reconstructed by 3D-SSRecNet and 3D-ReconstNet. Pixel2point has too many parameters to be executed, but it successfully generated point clouds after incorporating the Part2Point framework. When the resolution reached 9216 and 13,824, all three generators were initially unable to train and reconstruct point clouds due to excessively high model parameters. However, adopting Part2Point, 3D-SSRecNet, and 3D-ReconstNet can successfully generate point clouds.

Overall, the previous experimental results show that the proposed framework in this paper effectively enhances the resolution and part details of generated point clouds with lower computational costs and memory demands.

## 5. Conclusions and Future Work

High-resolution 3D modeling is critical to the development of VR/AR and computer vision applications. When the 3D reconstruction models proposed in past research are used to generate high-resolution object models, they often face the problem of high memory demand and computation cost and cannot even successfully work because their parameter numbers extremely increase with model resolution. In this study, we have successfully developed a part-oriented point cloud generation framework called Part2Point to resolve this problem. This innovative framework incorporates object part segmentation to reconstruct point cloud models based on the local structure of objects. Through our experiments, we have demonstrated several key findings. First, Part2Point effectively reduces the model parameters of the point cloud generator architecture while achieving the same output resolution. This approach eliminates the need for many fully connected layer parameters in the decoder stage of the reconstruction network to output high-resolution point clouds. Second, segmenting objects into individual parts allows us to reconstruct point cloud models for each part separately rather than reconstructing the entire high-resolution point cloud simultaneously. This results in higher point cloud resolutions at a lower computational cost and reduced memory demand while simultaneously enhancing reconstruction accuracy by focusing on local structure. Third, increasing the number of segmented parts with the same point cloud resolution leads to an overall increase in point cloud resolution. This approach preserves more object details than previous methods that down-sample point clouds to 2048 points, especially for dense point clouds with higher resolutions or complex shapes. Fourth, part segmentation introduces computational time costs due to increased batches fed into the generator. However, it also reduces the model parameters, ultimately reducing point cloud reconstruction time, particularly as the resolution increases. Fifth, the part-oriented reconstruction framework enables neural networks to focus on learning object local structures, resulting in better point cloud reconstruction results, especially for capturing object details.

Our experimental result shows that when the resolution reaches 9216 and 13,824, the reconstruction quality improvement induced by part segmentation is insignificant. In the future, we will adjust the architecture of the point cloud generators with part segmentation to achieve better high-resolution reconstruction quality and evaluate the impact of the proposed framework on other 3D reconstruction models. On the other hand, we will address the time and resource-intensive process of manually annotating 2D and 3D parts by exploring unsupervised or semisupervised learning methods to generate part-annotation datasets. We also investigate using small amounts of unlabeled or partially labeled data to improve part segmentation, resulting in more annotated data and robust training of point cloud generator models. In addition, we will explore more efficient and discriminative loss functions to alleviate computational complexity challenges associated with the Earth mover’s distance (EMD) as point cloud resolution increases.

## Figures and Tables

**Figure 1 sensors-24-00034-f001:**
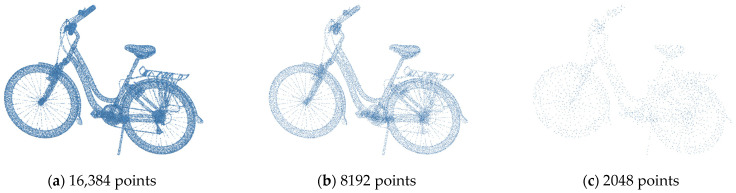
Resolution comparison of point clouds.

**Figure 2 sensors-24-00034-f002:**
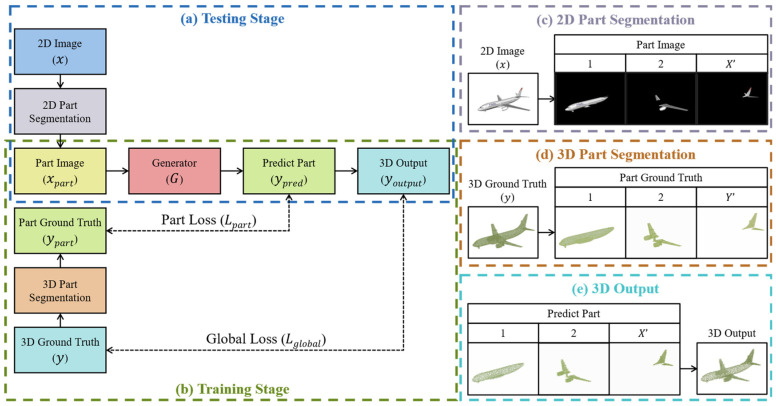
Framework of Part2Point.

**Figure 3 sensors-24-00034-f003:**
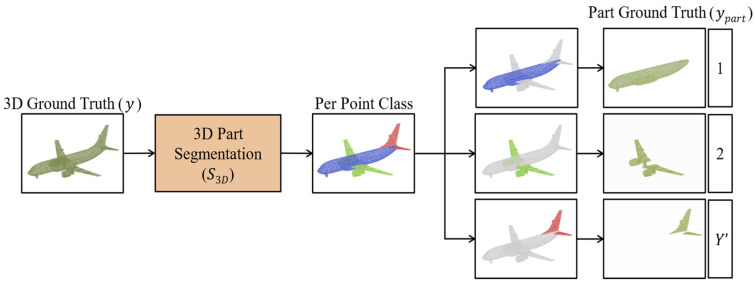
Three-dimensional part segmentation.

**Figure 4 sensors-24-00034-f004:**
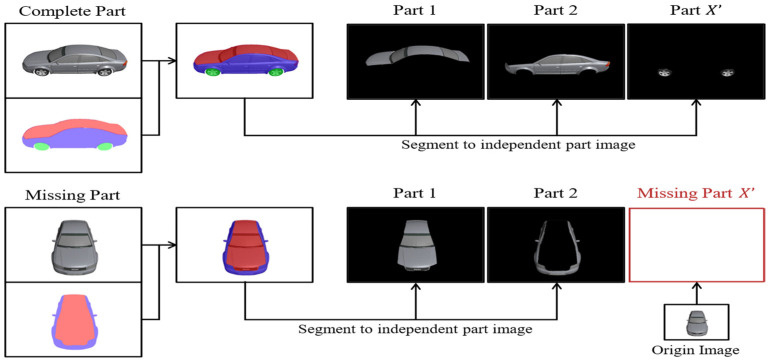
Two-dimensional part segmentation.

**Figure 5 sensors-24-00034-f005:**
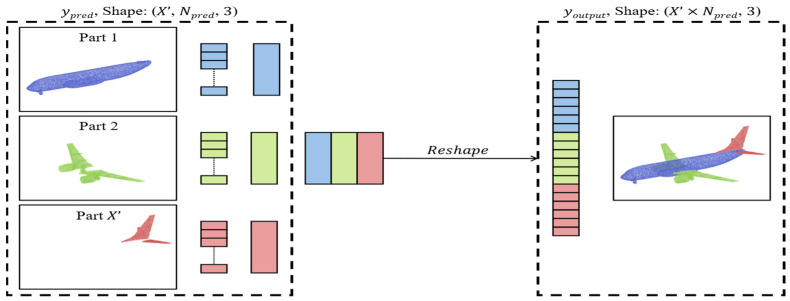
Merging part point clouds.

**Figure 6 sensors-24-00034-f006:**
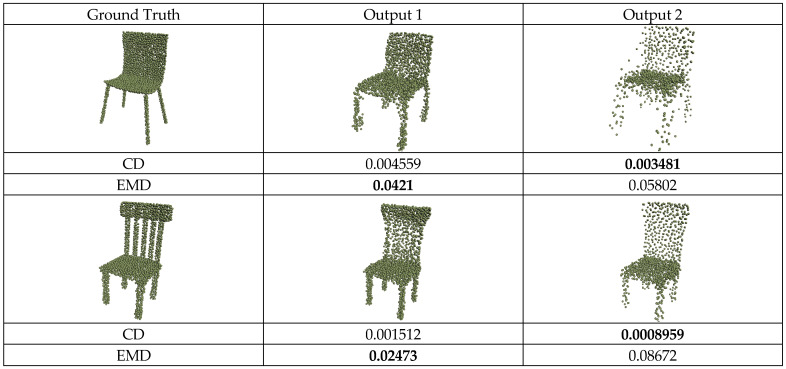
Visual comparison between CD and EMD.

**Figure 7 sensors-24-00034-f007:**
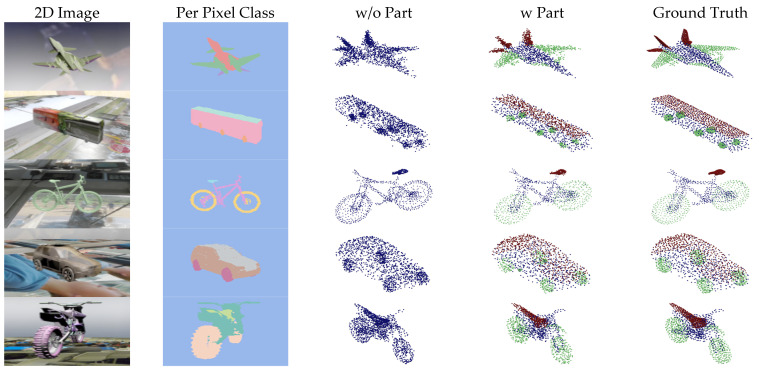
Visualization of point cloud reconstruction at a resolution of 1536.

**Figure 8 sensors-24-00034-f008:**
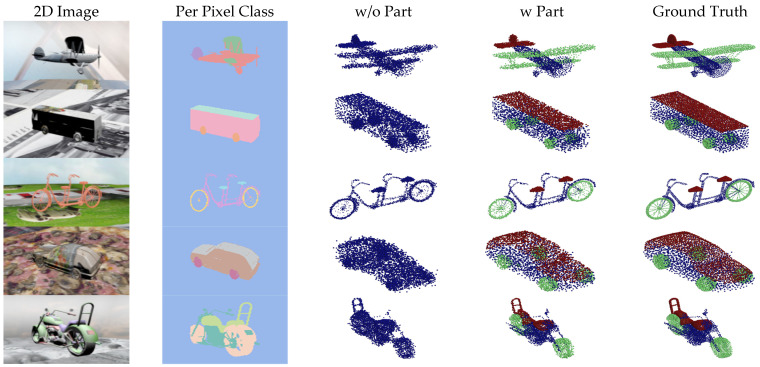
Visualization of point cloud reconstruction at a resolution of 4608.

**Figure 9 sensors-24-00034-f009:**
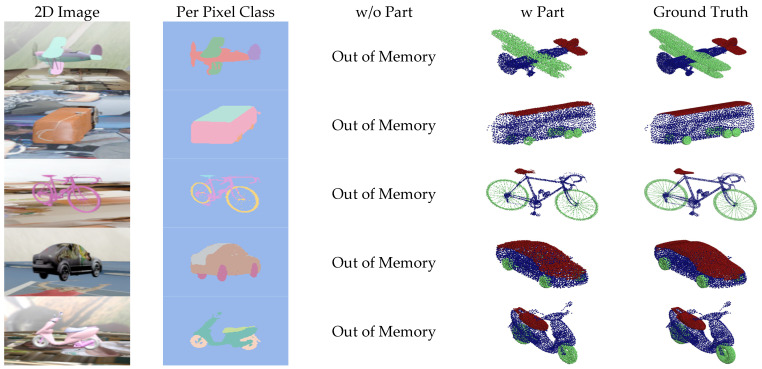
Visualization of point cloud reconstruction at a resolution of 9216.

**Figure 10 sensors-24-00034-f010:**
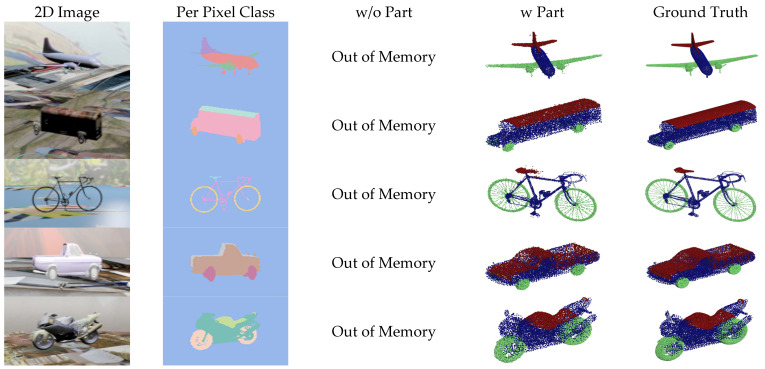
Visualization of point cloud reconstruction at a resolution of 13,824.

**Table 1 sensors-24-00034-t001:** Comparison of related work.

Research	Year	Encoder	Feature Vector	Decoder	Image Process	Reconstruct Target	Resolution	Approach
3D-ReconstNet [27]	May 2020	ResNet50	1 × 100	FC	Normalization	Global Point Cloud	1024 2048	Using VAE methods to produce plausible point cloud results for blurry or slurred images.
Pixel2point [28]	December 2020	Conv	256 × 259	FC	Normalization	Global Point Cloud	2048	Using the spherical point cloud template with the features of the image feature extraction network.
Yuniarti et al. [29]	November 2021	ResNet50 PointNet++	2048 (2D:1024) (3D:1024)	Conv	Normalization	Global Point Cloud	1024	Different input images using different point cloud templates to reconstruct.
Ping et al. [30]	August 2021	Conv	512	FC	Normalization	Global Point Cloud	1024	Generating edges and corners of the image, and reprojecting the output point cloud back to the plane.
Liu et al. [31]	October 2021	Conv	Nx2435	resGraphX (FC + GCNN)	Normalization	Global Point Cloud	1024	Reconstructing point clouds using random point clouds and multiscale features.
Chen et al. [32]	December 2021	Conv PointNet	1024	Linear	Normalization	Global Point Cloud	1024	Searching for real point clouds of similar objects in the database to aid reconstruction.
3D-SSRecNet [33]	October 2022	DetNet	1 × 100	FC	Normalization	Global Point Cloud	1024 2048	Image feature extractor based on the DetNet backbone network to bring a higher perception field.
Part2Point	July 2023	ResNet50	1 × 100	FC	Normalization Gamma Adjust	Part Point Cloud	1536 4608 9216 13,824	Using part segmentation to reconstruct the local parts of the object separately to improve the reconstruction resolution and quality.
		Conv	256 × 259	FC				
		DetNet	1 × 100	FC				

**Table 2 sensors-24-00034-t002:** Impact of part segmentation on parameter number and training time.

Baseline	Resolution	Parameters × 10^6^		Batch Time (ms)	
		w/o Part	w Part	w/o Part	w Part
3D-ReconstNet	1536	34.0985	**26.7026**	**55.5416**	81.4390
	4608	100.3262	**34.0985**	226.3058	**133.8352**
	9216	*	**58.9548**	*	**322.4979**
	13,824	*	**100.3262**	*	**639.8990**
Pixel2point	1536	645.8794	**186.8478**	99.1192	**43.0830**
	4608	*	**645.8794**	*	**150.4784**
	9216	*	*	*	*
	13,824	*	*	*	*
3D-SSRecNet	1536	26.8140	**19.4182**	**56.3019**	87.7139
	4608	93.0417	**26.8140**	227.1828	**140.5757**
	9216	*	**51.6703**	*	**329.6051**
	13,824	*	**93.0417**	*	**646.7262**

* GPU out of memory.

**Table 3 sensors-24-00034-t003:** Impact of part segmentation on reconstruction errors.

Baseline	Resolution	EMD × 10^−3^	
		w/o Part	w Part
3D-ReconstNet	1536	**3.0150**	3.6341
	4608	2.7259	**2.4058**
	9216	*	**2.7349**
	13,824	*	**2.9141**
Pixel2point	1536	**4.5801**	4.7705
	4608	*	**3.4979**
	9216	*	*
	13,824	*	*
3D-SSRecNet	1536	3.9524	**3.6830**
	4608	2.8710	**2.7549**
	9216	*	**2.7639**
	13,824	*	**3.0988**

* GPU out of memory.

**Table 4 sensors-24-00034-t004:** Impact of part number on parameter number and training time.

Baseline with Part2point	Resolution	Parameters × 10^6^		Batch Time (ms)	
		Part 2	Part 3	Part 2	Part 3
3D-ReconstNet	1536	27.8635	**26.7026**	**67.7732**	81.4390
	4608	44.4623	**34.0985**	155.3247	**133.8352**
	9216	100.3262	**58.9548**	431.5171	**322.4979**
	13,824	*	**100.3262**	*	**639.8990**
Pixel2point	1536	290.3990	**186.8478**	56.9126	**43.0830**
	4608	*	**645.8794**	*	**150.4784**
	9216	*	*	*	*
	13,824	*	*	*	*
3D-SSRecNet	1536	20.5790	**19.4182**	**70.7718**	87.7139
	4608	37.1778	**26.8140**	158.5024	**140.5757**
	9216	93.0417	**51.6703**	435.3806	**329.6051**
	13,824	*	**93.0417**	*	**646.7262**

* GPU out of memory.

**Table 5 sensors-24-00034-t005:** Impact of part number on reconstruction errors.

Baseline with Part2point	Resolution	EMD × 10^−3^	
		Part 2	Part 3
3D-ReconstNet	1536	**3.2090**	3.6341
	4608	2.6103	**2.4058**
	9216	3.3004	**2.7349**
	13,824	*	**2.9141**
Pixel2point	1536	**4.5668**	4.7705
	4608	*	**3.4979**
	9216	*	*
	13,824	*	*
3D-SSRecNet	1536	**3.2416**	3.6830
	4608	2.8234	**2.7549**
	9216	2.8905	**2.7639**
	13,824	*	**3.0988**

* GPU out of memory.

**Table 6 sensors-24-00034-t006:** Impact of global loss on reconstruction errors.

Baseline with Part2point	Resolution	EMD × 10^−3^	
		w/o Global Loss	w Global Loss
3D-ReconstNet	1536	5.5664	**3.6341**
	4608	5.1782	**2.4058**
	9216	5.2911	**2.7349**
	13,824	5.7327	**2.9141**
Pixel2point	1536	5.4261	**4.7705**
	4608	4.9779	**3.4979**
	9216	*	*
	13,824	*	*
3D-SSRecNet	1536	5.5236	**3.6830**
	4608	5.8300	**2.7549**
	9216	5.8053	**2.7639**
	13,824	5.7876	**3.0988**

* GPU out of memory.

**Table 7 sensors-24-00034-t007:** Reconstruction errors for different object categories.

Baseline	Class	1536		4608		9216		13,824	
		EMD × 10^−3^		EMD × 10^−3^		EMD × 10^−3^		EMD × 10^−3^	
		w/o Part	w Part	w/o Part	w Part	w/o Part	w Part	w/o Part	w Part
3D-ReconstNet	airplane	**2.2942**	3.0801	2.4486	**2.3944**	*	**2.4032**	*	**2.6453**
	bus	**2.9452**	4.2946	**3.1775**	3.9811	*	**3.6653**	*	**3.8379**
	bicycle	4.3406	**2.1741**	4.6802	**1.8245**	*	**1.8001**	*	**2.7155**
	motorbike	3.1908	**3.1580**	3.3843	**2.9898**	*	**2.8782**	*	**4.0297**
	car	**3.1127**	5.0973	**1.9555**	3.4091	*	**2.9618**	*	**4.0415**
Pixel2point	airplane	**3.5436**	3.5937	*	**3.2848**	*	*	*	*
	bus	**4.1336**	6.1725	*	**4.0799**	*	*	*	*
	bicycle	7.9504	**3.3159**	*	**2.2445**	*	*	*	*
	motorbike	**3.5840**	4.4104	*	**3.4684**	*	*	*	*
	car	**3.2411**	6.1725	*	**3.4585**	*	*	*	*
3D-SSRecNet	airplane	**2.3918**	2.5436	2.2183	**2.2052**	*	**2.2142**	*	**3.2820**
	bus	**3.1402**	3.8450	**2.8995**	2.9512	*	**3.3192**	*	**4.9274**
	bicycle	6.7063	**2.3678**	4.3667	**2.2045**	*	**2.1010**	*	**5.4872**
	motorbike	3.5998	**3.5922**	3.5251	**2.7118**	*	**2.3621**	*	**3.3876**
	car	**3.2382**	4.6990	**1.9924**	2.8603	*	**2.8605**	*	**4.2428**

* GPU Out of Memory.

## Data Availability

The UDA-Part dataset is publicly available online. The public dataset can be found at https://qliu24.github.io/udapart. The ShapeNet dataset is available at https://shapenet.org/.

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
