# Peer review of "Part2Point: A Part-Oriented Point Cloud Reconstruction Framework"

_sensors, 2023, doi:10.3390/s24010034_

Round 1

Reviewer 1 Report

Comments and Suggestions for Authors

The paper is very well organized and presented. The topic is trending in the 3D reconstruction communities and with good reason. I have no observations to make, everything looks good. But there is a 'but'.

In the introduction, it is stated that in the past creating a 3D model took much time, and then the authors exemplify with 3D modeling and 3D laser scanning. The first they say it is laborious while the second is too expensive and limited to case specificity, which is true. But they don't mention SfM algorithms. Photogrammetry provides the most realistic results today and is also based on 3D data extraction from 2D images. I believe a discussion on this topic must also include photogrammetry.

Excluding the big miss with photogrammetry from the arguments, the Introduction section is well structured and argumented from the new machine learning algorithms for 3D data extraction from 2D images point of view.

A comprehensive review/presentation of previous and related works with a good comparison.

The method is well explained, easy to follow, and should be understood even by non-specialists.

The results and conclusions are clearly and transparently presented. 

Author Response

Please also see the attachment file.

Response for the review comments

Comment 1)

In the introduction, it is stated that in the past creating a 3D model took much time, and then the authors exemplify with 3D modeling and 3D laser scanning. The first they say it is laborious while the second is too expensive and limited to case specificity, which is true. But they don't mention SfM algorithms. Photogrammetry provides the most realistic results today and is also based on 3D data extraction from 2D images. I believe a discussion on this topic must also include photogrammetry.

Response) Thanks for the reviewer’s suggestion. We have discussed the SfM algorithm in the introduction section of our manuscript, as follows.

In Page 2

However, creating 3D models took much work in the past. Application developers had to manually use computer-aided design software such as Blender [6], Solidwork [7], and ProE [8] to model objects one by one according to their structures. To save labor and time, an alternative method is to scan physical objects using 3D scanners directly. However, the scanning range of 3D scanners is limited, and high-resolution 3D scanner equipment is expensive, making it difficult for the general public to afford. By contrast, structure-from-motion (SFM) photogrammetry [9][10][11] is a practical and realistic solution to create 3D models of physical objects from a collection of two-dimensional images or video frames. The steps involved in SfM mainly include feature detection and matching, camera pose estimation, triangulation, and point cloud generation. SfM is a well-established method with simplicity and versatility, while is sensitive to outliers in feature matching and relies on dense and distinctive features for accurate reconstruction. It is necessary to seek or integrate with an alternative approach for generating promising 3D reconstructions even with sparse input data and handling low-texture and occlusion environments more effectively by learning contextual information.

Reviewer 2 Report

Comments and Suggestions for Authors

In this study, a new point cloud reconstruction framework is proposed. However, the study needs to be revised in terms of novelty and writing language. Here are my suggestions:

1) English writing language should be improved. There are spelling mistakes and repetitive expressions. A more fluent language should be used.

2) Case studies should be increased by placing more emphasis on the feature extraction aspect of the methods presented in the "2.1 Feature extraction" section.

3) 3D semantic segmentation studies should be added in the “2.4 Semantic segmentation:” section. The added studies are generally used for image semantic segmentation. Associate semantic segmentation studies with the work you present. Additionally, more publications should be added for point cloud semantic segmentation. Some suggestions:

Atik, M.E.; Duran, Z. An Efficient Ensemble Deep Learning Approach for Semantic Point Cloud Segmentation Based on 3D Geometric Features and Range Images. Sensors 202222, 6210. https://doi.org/10.3390/s22166210

Liu, L.; Yu, J.; Tan, L.; Su, W.; Zhao, L.; Tao, W. Semantic Segmentation of 3D Point Cloud Based on Spatial Eight-Quadrant Kernel Convolution. Remote Sens. 202113, 3140. doi: 10.3390/rs13163140

5) When Table II is examined, it is seen that the proposed method has more parameters than other methods. This does not support the claim of fewer parameters in the abstract section. An explanation regarding this situation should be added.

6) Which data set the results belong to should be written in the tables and figures.

7) When the results are examined, it is seen that the error amount of the proposed method is not much different from other methods, and is even higher. At this stage, the necessity and originality of the study should be explained more clearly. More data sets can be added. Thus, its performance can be examined in different data sets. Additionally, different methods can be added for comparison.

8) The stages of the proposed method should be explained mathematically. A description of the evaluation metrics should also be included.

Comments on the Quality of English Language

English writing language should be improved. There are spelling mistakes and repetitive expressions. A more fluent language should be used.

Author Response

Please also see the attachment file.

Response to the review comments

Comment 1) English writing language should be improved. There are spelling mistakes and repetitive expressions. A more fluent language should be used.

Response) We have corrected spelling mistakes and had our words rephrased in our manuscript with the assistance of a native speaker.

Comment 2) Case studies should be increased by placing more emphasis on the feature extraction aspect of the methods presented in the "2.1 Feature extraction" section.

Response) We have introduced more feature extraction methods in the related work section as follows.

On Page 4:

  1. Engel et al. [23] proposed a deep learning network called Point Transformer to extract local and global features and relate both representations by the local-global attention mechanism to capture spatial point relations and shape information. This network is permutation invariant because of a module called SortNet that extracts ordered local feature sets from different subspaces. M.H. Guo et al. [24] presented Point Cloud Transformer (PCT) to capture better local features hidden in the point cloud by the offset-attention of implicit Laplace operator and the normalization mechanism that is inherently permutation-invariant and more effective than the original Transformer for learning point clouds. H. Zhao et al. [25] constructed a 3D point cloud understanding network based on point transformer layers, pointwise transformations, and pooling.

Comment 3) 3D semantic segmentation studies should be added in the “2.4 Semantic segmentation:” section. The added studies are generally used for image semantic segmentation. Associate semantic segmentation studies with the work you present. Additionally, more publications should be added for point cloud semantic segmentation. Some suggestions:

Response) According to the reviewer's suggestion, we have introduced more point cloud semantic segmentation publications in the related work section as follows.

On Page 5:

Atik et al.[51] present an effective ensemble deep learning method for semantically segmenting mobile LiDAR sensor point clouds. The approach involves projecting 3D mobile point clouds into a 2D representation using spherical projection on range images. Liu et al. [52] introduced the Spatial Eight-Quadrant Kernel Convolution (SEQKC) algorithm for enhancing 3D point cloud semantic segmentation, specifically targeting small objects in complex environments. The SEQKC algorithm improves the network's capability to extract detailed features, resulting in higher accuracy for small objects and boundary features.

Comment 5) When Table II is examined, it is seen that the proposed method has more parameters than other methods. This does not support the claim of fewer parameters in the abstract section. An explanation regarding this situation should be added.

Response) We highlighted the parameter number in Table II by the blue color. We have explained the experimental result as follows.

On Page 13:

Obviously, the parameters of the point-cloud generators with part segmentation are much less than those of the same generators without part segmentation. When part segmentation is utilized, the point cloud generators individually reconstruct point clouds for the part segments of a given object and then assemble the point clouds of part segments to form the complete point cloud of the object. For the same resolution, as the point cloud is divided into three parts, the number of output points of the decoder is only one-third of that without part segmentation. Consequently, the parameters of the generator models decrease significantly.

Comment 6) Which data set the results belong to should be written in the tables and figures.

Response) As described in the second paragraph of Section 4, we specifically mentioned what datasets were used in the tables and figures of our experimental results as follows.

On Page 12:

The datasets used in the tables and figures of experimental results are UDA-Part [50] and ShapeNet Part Annotation [53]. We utilized the multi-view 2D part segmentation dataset provided by UDA-Part and extracted corresponding 3D part segmentation point clouds from the ShapeNet Part Annotation as reconstruction targets for the 2D dataset. The dataset categories include cars, airplanes, bicycles, buses, and motorcycles.

7) When the results are examined, it is seen that the error amount of the proposed method is not much different from other methods, and is even higher. At this stage, the necessity and originality of the study should be explained more clearly. More data sets can be added. Thus, its performance can be examined in different data sets. Additionally, different methods can be added for comparison.

Response) Thanks for the reviewer’s suggestion. We have explained the result of Table II, and the necessity and originality of the study more clearly as follows.

On Page 14:

The results consistently show that employing part segmentation improves the EMD metric for point cloud reconstruction. This experiment highlights that integrating part segmentation into point cloud reconstruction tasks can enhance reconstruction results at most resolutions. Although the parameter number of the generator model is much less than the original, the generators with part segmentation can reduce the EMD values because they construct better point clouds for each part of the target object as they can focus on local part features. With increasing resolution, the model parameters also grow, enabling the model to capture finer object details, theoretically resulting in improved point cloud quality. However, the reconstruction results induced by 3D-ReconstNet and Pixel2point are not improved by part segmentation at the 1536 resolution. This result implies that although part segmentation can effectively reduce the model parameters necessary for reconstructing point clouds, the reconstruction accuracy may not be improved but degraded when the parameter number is reduced below a threshold. In addition, the improvement induced by part segmentation seems most pronounced when transitioning from 1536 to 4608 resolution. The possible explanation could be that the current model architectures for the generators are not yet optimized for higher resolutions like 9216 and 13824. Future work may require adjustments to the generator model architecture to achieve better reconstruction quality with more points.

On Page 25:

High-resolution 3D modeling is critical to the development of VR/AR and computer vision applications. When the 3D reconstruction models proposed in past research are used to generate high-resolution object models, they often face the problem of high memory demand and computation cost and cannot even successfully work because their parameter numbers extremely increase with model resolution. In this study, we have successfully developed a part-oriented point cloud generation framework called Part2Point to resolve this problem. This innovative framework incorporates object part segmentation to reconstruct point cloud models based on the local structure of objects. Through our experiments, we have demonstrated several key findings. First, Part2Point effectively reduces the model parameters of the point cloud generator architecture while achieving the same output resolution. This approach eliminates the need for many fully connected layer parameters in the decoder stage of the reconstruction network to output high-resolution point clouds. Second, segmenting objects into individual parts allows us to reconstruct point cloud models for each part separately rather than reconstructing the entire high-resolution point cloud simultaneously. This results in higher point cloud resolutions at a lower computational cost and reduced memory demand while simultaneously enhancing reconstruction accuracy by focusing on local structure. Third, increasing the number of segmented parts with the same point cloud resolution leads to an overall increase in point cloud resolution. This approach preserves more object details than previous methods that down-sample point clouds to 2048 points, especially for dense point clouds with higher resolutions or complex shapes. Fourth, part segmentation introduces computational time costs due to increased batches fed into the generator. However, it also reduces the model parameters, ultimately reducing point cloud reconstruction time, particularly as the resolution increases. Fifth, the part-oriented reconstruction framework enables neural networks to focus on learning object local structures, resulting in better point cloud reconstruction results, especially for capturing object details.

On the other hand, the proposed framework requires 2D and 3D part-annotation datasets for training and testing. However, most 3D part-annotation point cloud datasets do not provide paired 2D part-annotation images for their point clouds because manual multi-view labeling is time-consuming and vice versa. Therefore, we currently use only the 2D part-annotation dataset provided by UDA-Part and extracted corresponding 3D part-annotation point clouds from the ShapeNet Part Annotation as reconstruction targets for the 2D part-annotation dataset. Page 25 describes that our future work will address the time and resource-intensive process of manually annotating 2D and 3D parts by exploring unsupervised or semi-supervised learning methods to generate 2D-to-3D paired part-annotation datasets. In addition, we will evaluate the impact of part segmentation on other point cloud generation models.

8) The stages of the proposed method should be explained mathematically. A description of the evaluation metrics should also be included.

Response) We have mathematically expressed the stages of the proposed framework on Page 8 of our revised manuscript (as shown in the attachment file).

Round 2

Reviewer 2 Report

Comments and Suggestions for Authors

Thanks to the authors for the revisions. The paper is acceptable in this form.